# Combining the YOLOv4 Deep Learning Model with UAV Imagery Processing Technology in the Extraction and Quantization of Cracks in Bridges

**DOI:** 10.3390/s23052572

**Published:** 2023-02-25

**Authors:** Szu-Pyng Kao, Yung-Chen Chang, Feng-Liang Wang

**Affiliations:** Department of Civil Engineering, National Chung Hsing University, Taichung 40227, Taiwan

**Keywords:** bridge inspection, crack identification, deep learning, structural health monitoring, unmanned aerial vehicle

## Abstract

Bridges are often at risk due to the effects of natural disasters, such as earthquakes and typhoons. Bridge inspection assessments normally focus on cracks. However, numerous concrete structures with cracked surfaces are highly elevated or over water, and is not easily accessible to a bridge inspector. Furthermore, poor lighting under bridges and a complex visual background can hinder inspectors in their identification and measurement of cracks. In this study, cracks on bridge surfaces were photographed using a UAV-mounted camera. A YOLOv4 deep learning model was used to train a model for identifying cracks; the model was then employed in object detection. To perform the quantitative crack test, the images with identified cracks were first converted to grayscale images and then to binary images the using local thresholding method. Next, the two edge detection methods, Canny and morphological edge detectors were applied to the binary images to extract the edges of the cracks and obtain two types of crack edge images. Then, two scale methods, the planar marker method, and the total station measurement method, were used to calculate the actual size of the crack edge image. The results indicated that the model had an accuracy of 92%, with width measurements as precise as 0.22 mm. The proposed approach can thus enable bridge inspections and obtain objective and quantitative data.

## 1. Introduction

The detection of defects on the surface of concrete structures is a vital part of structural health monitoring. The detection of defects, such as cracks, exposed bars, and corrosion on the surface of bridges is necessary due to the effects of such defects on the durability of concrete structures [1]. Cracks are the most common and most consequential defect because they represent insufficient strength and a decrease in the safety of a bridge [2]. Crack width is a main criterion used to assess the performance of concrete components and is critical in ensuring bridge performance [3,4,5].

Cracks can be detected through manual visual inspection and assessment. Conventional detection methods involve a bridge inspector using an engineering vehicle or a water vehicle (Figure 1) and making judgements about deterioration on the basis of their personal experience; this process is both dangerous and subjective [6], as well as labor-intensive and time-consuming [7]. To overcome the challenges posed by this conventional method, objective and automated methods must be developed. To this end, computer vision techniques can be employed [8]. UAVs can easily fly close to bridge components that are difficult for people to access. Using the camera integrated on the UAV (UAV-mounted camera), the camera can be controlled by a remote control on the ground, and UAV imagery can be taken to detect bridge cracks.

In recent years, structural crack identification and detection technology based on computer vision (CV) has been gradually applied to civil engineering operations and maintenance [9,10,11]. Computer vision techniques can be used to identify cracks in concrete. Conventional machine-learning algorithms include linear regression, decision trees, the support vector machine, and the Bayesian algorithm. The disadvantage of computer vision is that it is affected by the presence of different objects, such as light, shadows, and rough surfaces [12]. However, various hybrid approaches of artificial intelligence (AI) and machine learning (ML) techniques can be used to overcome these limitations [13,14,15,16]. Sharma et al. (2018) combined a support vector machine with a convolutional neural network (CNN) to identify cracks in reinforced concrete; their method has a higher identification accuracy than does the use of a convolutional neural network alone [17]. Prateek et al. employed computer vision techniques to extract image features from images of cracks in concrete and then trained their system to extract information about the cracks [18]. This method can only identify information about spatial positions, gray values, and saturation; it cannot extract depth features, and its identification accuracy is low. Furthermore, machine learning does have weaknesses, such as slow convergence rates and large time requirements [19,20]. To achieve more rapid identification, automated feature extraction techniques must be established.

Deep learning is an important branch in the field of machine learning. In the field of deep learning, convolutional neural network (CNN) is the most prominent for image recognition. The convolutional neural network consists of convolutional layers, pooling layers, flattening layers, and fully connected layers, while the number of layers is not fixed. The number of layers in each model is different, so there are different results in the analysis. The vigorous development of deep learning in recent years has given rise to novel approaches for detecting cracks in concrete surfaces, often through the application of CNNs to extract crack features. CNNs can automatically learn depth features from training images and thus have high crack detection efficiency and accuracy [7,21]. Cha et al. (2017) proposed a deep CNN for sorting images on the basis of whether they show cracks [7]. Liang et al. (2017) used a deep CNN to classify concrete cracks and spalling; the challenge in their approach, which is based on image segmentation, is finding the appropriate sub-image size when cracks of various sizes are present [22]. Gang Yao et al. (2021) improved the YOLOv4 model to realize real-time detection of concrete surface cracks. The results show that the improved YOLOv4 model reduces parameters and calculations by 87.43% and 99.00%, respectively. Although the identification speed has been improved, the mAP of crack detection is 94.09%, which is slightly lower than the 98.52% of the original YOLOv4 model [23]. Qianyun Zhang et al. (2021), in order to improve the training efficiency, first transformed images into the frequency domain during a preprocessing phase. An integrated one-dimensional convolutional neural network (1D-CNN) and a long short-term memory (LSTM) method was used in the image frequency domain. The algorithm was trained using thousands of images of cracked and non-cracked concrete bridge decks. The accuracy of the developed model was 99.05%, 98.9%, and 99.25%, respectively, for training, validation, and testing data. The 1D-CNN-LSTM algorithm makes it a promising tool for real-time crack detection [24].

Target detection models can be distinguished into one-stage and two-stage models. The SSD [25] and YOLO [26] models are one-stage models that treat target detection as a regression problem. The region-based CNN (R-CNN) [27], fast R-CNN [28], faster R-CNN [29], and spatial pyramid pooling (SPP) network [30] are two-stage models. When training a two-stage model, the target region detection network is trained after the region proposal network (RPN) has been trained; consequently, two-stage models are highly accurate but slow. To complete an entire detection process and achieve end-to-end object detection without an RPN, initial anchors can be used in a one-stage model to predict classes and locate target regions. One-stage models are fast but relatively inaccurate. The accuracy and inference speed of target detection algorithms are key problems in the field of target detection, and the balance between efficiency and accuracy is a key technical issue. YOLOv4 satisfies these requirements because it has decent processing speed and performance [23]. Therefore, this study uses the YOLOv4 model as a model for identifying cracks.

Some studies [31,32,33] have verified the effectiveness of using image-processing techniques to extract information about cracks from images. Examples include the use of thresholding to convert cracks and their background into black and white pixels, edge detection techniques to extract the outlines of cracks, mathematical morphology to improve the overall shape of cracks in images, and Canny algorithms to minimize missed detection of crack edges [34,35]. Photographs are often affected by complex lighting conditions, shadows, and the randomness of the shape and size of cracks. Interference from weather conditions and brightness levels also lower the performance of concrete damage detection models [36,37].

Popular approaches for quantitative analysis of crack width involve the use of deep-learning and image-processing techniques. Kim et al. detected cracks on a bridge’s surface by using an R-CNN and then employed planar markers to quantitatively analyze the cracks; the crack width measurements were discovered to be precise down to 0.53 mm [38]. Park et al. employed the YOLOv3 model and structured light to identify and quantify crack features. To prevent data being affected by laser beam installation and manufacturing errors, the position of the laser beam was calculated and calibrated using a laser distance sensor. The thinnest crack that could be measured had a width of 0.91 mm [39]. Kim et al. (2019) proposed a mask R-CNN for detecting cracks and used morphological operations to quantify the crack width. The results indicated that cracks that were wider than 0.3 mm could be successfully measured, with the errors being smaller than 0.1 mm; however, the error was larger when cracks had a width of less than 0.3 mm [40].

The following shortcomings were identified in the reviewed studies: (1) the accessibility of deteriorated components must be considered when placing planar markers, (2) cracks are generally measured as wider than they actually are, and (3) the installation of laser detection systems on an unmanned aerial vehicle (UAV) requires additional system calibration. In the present study, a model for effectively identifying cracks under uneven lighting and given complex component backgrounds was trained using YOLOv4. Because most concrete bridges either span a river or are an elevated expressway, planar markers cannot be placed on them easily. In these situations, a total station can be employed to measure the coordinates of features on the concrete surface, and these coordinates can then be employed to calculate spatial distances as an alternative to planar markers. Research has verified the effectiveness of this approach when markers cannot be placed on a bridge. Furthermore, the proposed approach does not require a UAV to carry a ranging system or be modified and can measure crack widths smaller than 0.22 mm.

## 2. Materials and Methods

### 2.1. Research Method

Crack measurements in this study were performed using the scale method. Once the planar markers were placed next to a crack, a total station was set up to measure the wall features. These two methods were the basis for measuring the crack width in images. Smartphones, camera-equipped UAVs, and open-source data on cracks (SDNET 2018 dataset) were employed as the data for training the crack identification model. When conducting image identification tests with the trained model, thresholding and edge detection were used to extract the outlines of the cracks in images that had been identified as containing a crack. The widths of the extracted outlines were measured and compared with the actual values to determine the accuracy of the proposed model and to verify the feasibility of the proposed method (as depicted in Figure 2). The UAV (DJI Mavic 2 Pro) used in this study has a volume of 322 × 242 × 84 mm, a weight of 907 g, and a maximum flight time of 31 min. The situation of the UAV bridge detection operation is shown in Figure 3a. The UAV-mounted camera is shown in Figure 3b.

### 2.2. Attach the Planar Marker and Measurement Feature Points

The scale method was used to measure the widths of crack in images. Consequently, the images captured using a UAV had to contain planar markers, as well as five arrow flags under the crack (Figure 4). Because most concrete bridges either span a river or are an elevated expressway, planar markers cannot easily be placed on them. Taking such circumstances into consideration, a total station was used to measure the coordinates of the features of the concrete surfaces, and these coordinates were then used to reverse-calculate the spatial distances as an alternative to planar markers (Figure 5). Since the true distance of the planar marker or feature points in the image is known, the scale parameter for converting the image to the true size can be calculated, which is the scale method.

### 2.3. Crack Images

The acquisition of training data is a major step in model training. Images of cracks were obtained from a smartphone, a camera mounted on a UAV, and the deep-learning open-source dataset SDNET2018. The smartphone and UAV-mounted camera employed in this study are detailed in Table 1.

SDNET2018 is an annotated image dataset for training, validating, and benchmark testing AI-based concrete crack detection algorithms, and contains more than 56,000 images of concrete bridge surfaces, walls, and pavements with and without cracks. These images were 256 × 256 pixels in size(96 dpi), and the widths of the depicted cracks range from 0.06 to 25 mm (Table 2). The dataset also comprises images showing various obstructions, such as shadows, surface roughness, scaling, edges, holes, and background debris. Some images are not sufficiently zoomed in to show cracks clearly. This dataset was used with an AlexNet deep CNN framework to sort the captured images in accordance with whether cracks were depicted [21].

### 2.4. Crack Identification Model Training

Crack identification was performed using an object detection model comprising four parts: the input, the backbone, the neck, and the head. The input was the image that was entered into the model, whereas the backbone provided support for the training network and enabled feature extraction. YOLOv4 was developed by introducing the cross-stage partial network (CSPNet) into the Darknet53 architecture in YOLOv3; the resulting CSPDarkent53 architecture can be used to robustly acquire data from target objects. The neck comprised SPP and a path aggregation network (PANet); after features had been extracted in CSPDarknet53, they were condensed on multiple scales using pooling kernels in the SPP and then connected to each other through PANet. Once the features had been connected, vector features of fixed sizes could be obtained; these features ensured the model’s perception in object identification and lowered the computational load. Once the features had been integrated in the neck, the head layer predicted the bounding box coordinates, coordinate score, and classification label [43]. The biggest difference between a one-stage model and two-stage model is in the head; a one-stage model utilizes dense prediction, whereas a two-stage model involves sparse prediction [43]. In a one-stage model, numerous anchors are arranged on the image, and the possible target objects in the image are predicted through regression; consequently, the prediction outcomes are dense. In a two-stage model, a certain number of regions of interest are selected through a select search, leading to sparser prediction outcomes than for a one-stage model (Figure 6). 

To prevent training difficulties caused by insufficient training data, transfer learning was employed as the preliminary step in model training. When tagged data were insufficient, features could be extracted from another big database containing immense tagged data [38]. The MS COCO dataset was the basis for training through transfer learning in this study; it is a large image dataset provided by Microsoft, Facebook, and other organizations, and contains 330,000 images and 1.5 million objects across 80 categories. The COCO dataset was used as an initial model and the foundation for training the object detection model.

YOLO is an end-to-end one-stage object detection model that can generate identification outcomes immediately for an image input. The identification process is illustrated in Figure 7. The first step was the image input: the input image had to be in RGB format and the image’s dimensions had to be in multiples of 32 to be a YOLO-compatible format. The preset format is typically 416 × 416 [23], but this study uses 256 × 256, when performing crack detection using a YOLO model. Once an image was input, it was passed through the first object, the CBM, which stands for a convolution layer (C), batch normalization (B), and activation function Mish (M). The CBM was then split into several CSP networks, which are expressed as CSPX to indicate how many layers of neural networks they contain; the networks, in order, were CSP1, CSP2, CSP8, CSP8, and CSP4. Thus, the feature extraction portion was completed. The extracted data were then exported through three convolution layers to the neck, which comprised an SPP network and PANet. SPP kernels of size 13, 9, and 5 were used to condense the extracted features and export the multiscale feature maps. The feature maps were connected and underwent three more convolutions, thus completing the SPP portion. Then, the feature map was replicated and divided into two parts. Upsampling was conducted in one of the two parts, and the results of upsampling were merged with the output features of the second CSP8 in the CSPDarknet53 architecture; after five layers of convolution, upsampling was performed again. The upsampling results were merged with the output of the first CSP8 in the backbone, and after five convolutions, the features were replicated to produce two feature maps. One map was the YOLO head output. The second map was then combined with the feature outputs of the second CSP8 to perform downsampling, producing two feature maps. One of the feature maps was another YOLO head output, and the second map was combined with the second SPP output map and underwent five convolutions to yield the final YOLO head. The sizes of the three YOLO heads were 32, 16, and 8, and the dense prediction of objects was performed in the head.

### 2.5. Crack Identification and Measurement

Most object detection models use bounding boxes to frame possible objects within images. The framed range is cropped to the bounding box and then exported, and thresholding and edge detection are performed on the output to extract the outline of the crack. The width of the crack’s outline is then measured.

#### 2.5.1. Cropping to the Bounding Box

Figure 8 illustrates an identification outcome of the trained identification model. The black line in the middle of the figure represents the crack, and the black frames are the bounding boxes used in object detection. The number of bounding boxes was determined through regressive generation, and bounding boxes in an image were related to the number of image annotations. Figure 9 depicts the ideal detection method. In reality, most object detection models are unable to frame all of the cracks in an image because of some omissions or judgment errors in regressive-generated bounding boxes.

Figure 9 presents a schematic of one of the bounding boxes in Figure 8, and demonstrates that a bounding box provided image coordinates on the basis that the upper left corner of the image is the origin. The bounding box range could be extracted and image processing performed.

#### 2.5.2. Thresholding

Thresholding involves sorting an image into foreground and background pixels (0 and 1) in accordance with a given threshold value. The various approaches to thresholding can be divided into global or local thresholding. The main representative of global thresholding is Otsu’s method [44], whereas the most common form of local thresholding is the Sauvola method [45], in which the threshold calculation is adjusted by considering the distribution of values in regions adjacent to the pixels, as opposed to one threshold being selected for the whole image. Figure 10 describes the thresholding process, whereby the original image is converted into the grayscale image and then into the binarized image. The Sauvola thresholding formula used in this study is as follows:(1)$$T\left(x,y\right)=m\left[1+k\left(\frac{s\left(x,y\right)}{R}-1\right)\right]$$

Here, 

*T*: Obtained the threshold;

*m*: Mean grayscale intensity of the pixels within a custom-sized window;

*s*(*x*,*y*): Standard deviation of the grayscale intensity within a custom-sized window;

*R*: Dynamic range of the standard deviation;

*k*: Sensitivity parameter that controls the effects of the standard deviation.

#### 2.5.3. Crack Width Measurement

Following thresholding, the Canny edge detection and morphological edge detection were employed to extract the shapes of the cracks. Crack width in terms of pixels was calculated by aligning a crack vertically (Figure 11) and using Equation (2). As illustrated in Figure 12, when calculating crack width using Equation (2), the image had to contain a crack and planar markers; the actual dimensions of the crack could then be calculated in accordance with the actual and pixel dimensions of the markers and the pixel values of the crack.
(2)$$W_{r}=W_{pc}*\frac{l}{W_{ps}}$$

Here,

$W_{r}$ is the real crack width in metric units (mm);

$W_{pc}$ is the obtained crack width in pixels;

$W_{ps}$ is the planar marker width in pixels;

*l* is the planar marker width in metric units (mm).

## 3. Experiment Analysis

### 3.1. Outdoor Bridge Inspection Tests

The training process was implemented on a server with a high-performance GPU (NVIDIA GeForce GTX 1060), 64 GB DDR4 memory, and an Intel(R) Core™ i7-8700K CPU. The training process is based on the deep learning framework Darknet. The frame rate of the system recognition image is 30 fps. The training images in this study were photographs obtained using a smartphone camera of cracks on Xihu River Bridge, Houlongguanhai Bridge, and Touwu Bridge in Taiwan. A total of 379 photographs, measuring 1108 × 1478 pixels, were captured at a distance of 50–70 cm. These images were spliced into 256 × 256 images, resulting in a total of 1463 images. From the SDNET2018 dataset, 3006 images were selected after eliminating those that were too blurry or contained too much noise. Once all of the crack images had been amplified, cropped, cleaned, and annotated, they were assigned to the training dataset or testing dataset in an 8:2 ratio (Table 3). In addition, photographs of the cracks in concrete structures along Fuxing Rd in Taichung were captured using a UAV from a distance of 1 m. Once they were segmented and amplified, the images were used as testing images with uneven lighting and a complex background.

The aerial photographs were taken with a DJI UAV. Because the camera mounted on the UAV was not a range camera, it was affected by image system errors (such as poor radial distortion). At present, many scholars employ images of black and white checkerboards in photographs and then calculate the calibration parameters using a program [24]. In this study, a MATLAB camera calibrator was used to calibrate the camera and correct for any geometric deformation in the images (Table 4).

### 3.2. Results of the Crack Identification Model Training 

Deep-learning-based object detection involves measurement through objective evaluation indicators, which are various and include precision, recall, intersection over union (IOU), and mean average precision (mAP) [15]. The IOU is the proportion of the intersection of the bounding box predicted by the model and the actual bounding box divided by the union of both; it is also known as the Jaccard index. The mAP is an indicator of accuracy in object detection [15] and was therefore used in this study as a metric for evaluating the bridge crack identification model. The identification results of the YOLOv4-trained model are presented in Figure 13, which shows that the mAP reached 92%.

### 3.3. Crack Image Identification

The image identification outcomes could be divided into images with different lighting and background noise (Table 5), and the crack identification results are shown in Figure 14. Referencing the research conducted by Jiang et al. [27], in tests involving uneven lighting, crack images were sorted into three lighting condition groups in accordance with their range of pixel brightness: insufficient light (0 ≤ pixel brightness ≤ 80), adequate light (81 ≤ pixel brightness ≤ 160), and strong light (161 ≤ pixel brightness ≤ 255). Pixel brightness was calculated by converting an image to grayscale and then calculating the mean pixel value of the whole grayscale image. Images with a simple background were compared with those with a complex background under various lighting conditions to obtain the model’s identification outcomes.

The crack identification results obtained by the trained model and illustrated in Figure 14 clearly demonstrate that the proposed model could identify cracks in images. The positions of the bounding boxes were regressively determined; thus, not every predicted box could be guaranteed to frame a crack accurately and precisely. However, the results obtained in this study revealed that this did not lead to judgment errors for most images. Three bounding boxes may have been generated because when the image annotation files were created, in each image, the number of frames per crack was mostly three. This phenomenon reflects the importance of the training dataset and indicates that how cracks are tagged affects how results are displayed.

### 3.4. Quantitative Crack Analysis

In the identified crack image presented in Figure 14, the subject of interest is the crack within the bounding boxes. To quantitatively analyze this crack, the rectangular areas were cropped and thresheld to separate the pixels into foreground and background pixels. Edge detection was then performed to extract the outline of the crack. The detailed steps are as follows.

#### 3.4.1. Bounding Box Cropping

Figure 15 presents a screenshot of the coordinates and parameters of the bounding boxes predicted in Figure 16. The percentage is the confidence of being identified as a crack, within the parentheses, left_x is the horizontal displacement of the upper-left corner of the box from the origin, and top_y is the vertical displacement from the same; width and height are the dimensions of the bounding box. A Python program was written that cropped the images to within the three bounding boxes in accordance with the coordinates and parameters; the subsequent thresholding and width measurements were based on these cropped images.

#### 3.4.2. Crack Width Measurement

Once the pixels showing the crack had been separated into foreground and background pixels using the Sauvola local thresholding method (Figure 17), edge detection was performed to obtain the outline of the crack and facilitate the quantitative analysis. Canny and morphological edge detection were compared; the results are displayed in Figure 18 and Figure 19. The effects of the two edge detection methods are very similar. In order to analyze which one has the best effect, it is necessary to further convert the image width from pixel to mm, and then compare it with the true width of the crack for quantitative analysis.

#### 3.4.3. Spatial Resolution of the Single-Pixel Scale Method

The office and field scale methods employed in this study resulted in similar degrees of accuracy for both indoor and outdoor experiments. Consequently, during the bridge inspections, planar markers can be replaced with a total station when bridge crack detection is performed using the scale method.
$$\mathrm{Planar}\;\mathrm{marker}\;\mathrm{spatial}\;\mathrm{resolution}=\frac{\mathrm{Actual}\;\mathrm{length}\;\mathrm{of}\;\mathrm{planar}\;\mathrm{marker}}{\mathrm{Number}\;\mathrm{of}\;\mathrm{pixels}\;\mathrm{in}\;\mathrm{planar}\;\mathrm{marker}\;\mathrm{image}}=0.216\;\mathrm{mm}/\mathrm{px}$$
$$\mathrm{Total}\;\mathrm{station}\;\mathrm{spatial}\;\mathrm{resolution}=\frac{\mathrm{Actual}\;\mathrm{length}\;\mathrm{according}\;\mathrm{to}\;\mathrm{feature}\;\mathrm{point}\;\mathrm{coordinates}}{\mathrm{Number}\;\mathrm{of}\;\mathrm{pixels}\;\mathrm{in}\;\mathrm{feature}\;\mathrm{point}\;\mathrm{image}}=0.221\;\mathrm{mm}/\mathrm{px}$$

#### 3.4.4. Crack Width Quantification Results

The planar marking and total station scale methods were compared for the measurements made in the Canny and morphological edge detection images; the results are presented in Table 6, Table 7, Table 8 and Table 9. Because the differences between the two methods were discovered to be small, the accuracy of the crack width measurements was determined for the edge detection technique. The crack widths listed in Table 6 and Table 8 are identical to those listed in Table 7 and Table 9, except for a 1-pixel difference for Crack 3. The morphological edge detection was found to outperform the Canny edge detection. The different values displayed in Table 7 and Table 9 are speculated to be errors caused by the assignation of a threshold value in the Canny method; morphological edge detection does not involve the assignation of a threshold value; instead, the eroded part is subtracted from the foreground pixels. The results of the study show that the total station measurement method proposed in this study to measure the crack width has the same measurement accuracy as the planar marker method proposed by Kim et al. (2018) [30]. However, the disadvantage of the planar marker method is that if the detection target is too high, the planar marker cannot be attached close to it, but our method does not have this disadvantage and provides a more convenient bridge crack measurement method.

## 4. Conclusions

Using transfer learning, an object detection model (YOLO-v4 deep learning model) was trained and found to have an accuracy of 92%. The model performed favorably in the identification of cracks in images with uneven lighting and a complex background, proving that the model trained in this study had a good crack detection accuracy.

The research showed that the overall crack measurement accuracy was superior to 0.22 mm. The measurement performance of the two edge detection methods was similar. However, the Canny edge detector produces different crack edges when given different thresholds, which resulted in a more significant difference between the measured value and the actual value of the width; moreover, the morphological edge detector does not require the use of thresholds, and hence, it can produce crack edges close to the truth.

This study compared the conversion precisions on the two types of scale methods. The results showed that the difference between the two was only 0.005 mm. Personnel could not approach and affix the planar marker next to the bridge crack for viaducts or river crossings. The total station measurement method proposed in this study can achieve the same measurement accuracy as the planar marker method for measuring crack width. Hence, the method proposed in this study can eliminate the limitations of planar marker methods, providing a more convenient operating procedure for bridge crack size measurement.

It is recommended to improve this method by the following directions in subsequent studies: (1) Sauvola’s local thresholding method adopted in this study can convert grayscale images to binary images. We may test this method on images under different backgrounds or environmental conditions in future studies to find the corresponding optimal threshold values. (2) The trained model can be installed in the embedded system. Then, the embedded system can be integrated into the UAV body to realize real-time detection and measurement of bridge cracks. (3) A collection of more images with bridge defects can be used to extend the datasets and further improve the accuracy of the detection methods proposed in this study.

## Figures and Tables

**Figure 1 sensors-23-02572-f001:**
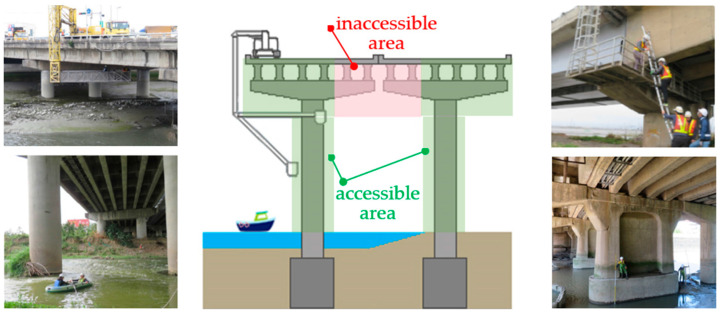
Traditional bridge inspection operations (bridge inspection equipment).

**Figure 2 sensors-23-02572-f002:**
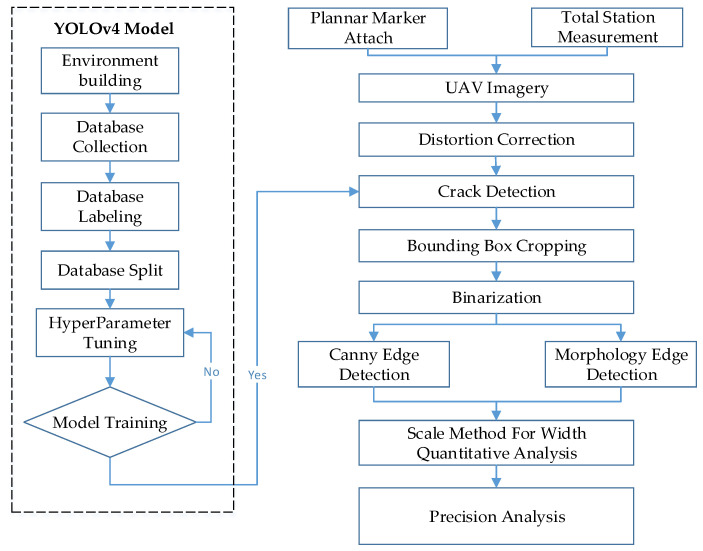
Research process.

**Figure 3 sensors-23-02572-f003:**
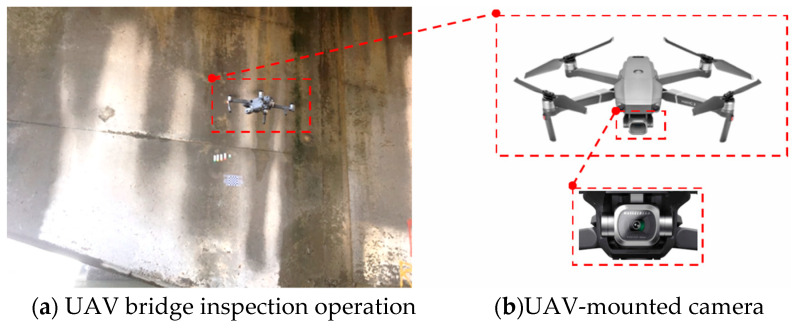
Diagram of the UAV bridge inspection operation.

**Figure 4 sensors-23-02572-f004:**
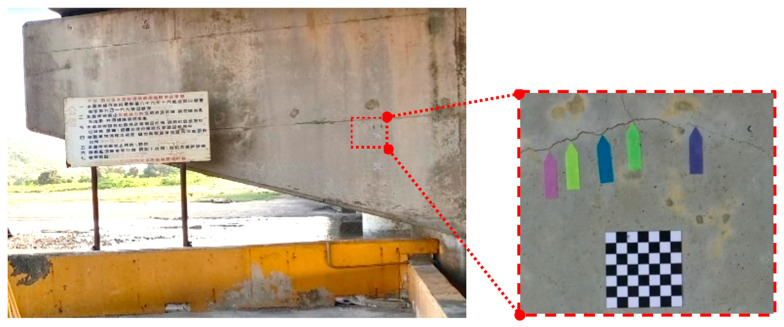
Affix the planar marker.

**Figure 5 sensors-23-02572-f005:**
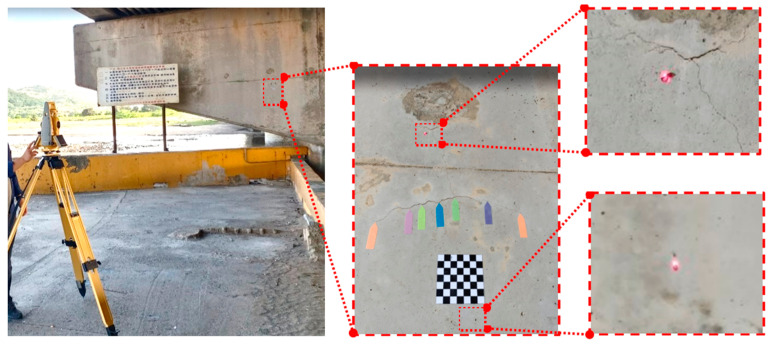
Total station measurement feature points.

**Figure 6 sensors-23-02572-f006:**
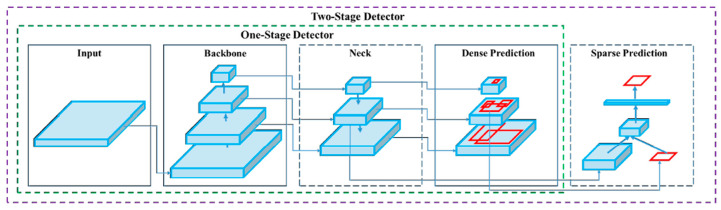
Architecture of the one-stage and two-stage object detectors [32].

**Figure 7 sensors-23-02572-f007:**
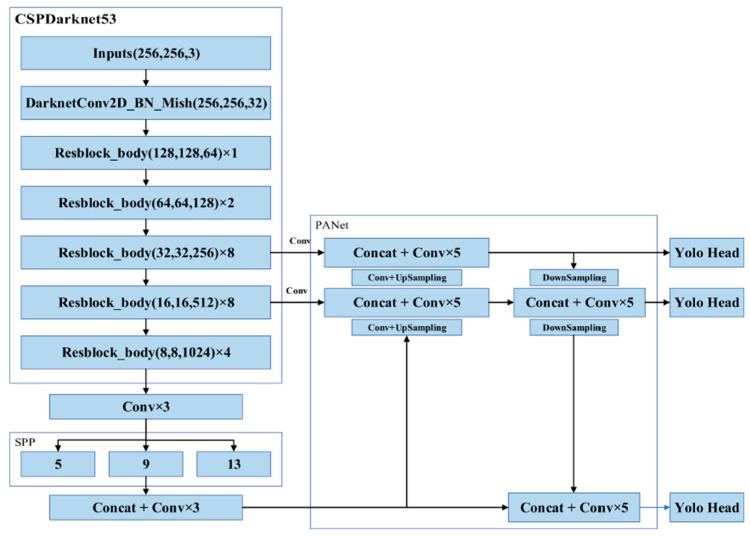
YOLOv4 model architecture [23].

**Figure 8 sensors-23-02572-f008:**
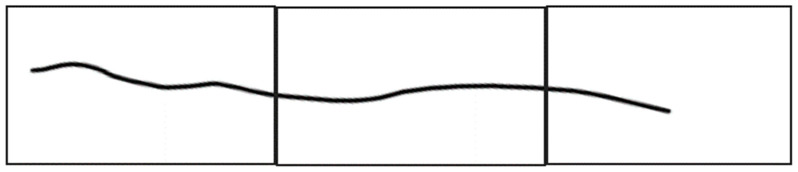
Bounding box detection of a crack.

**Figure 9 sensors-23-02572-f009:**
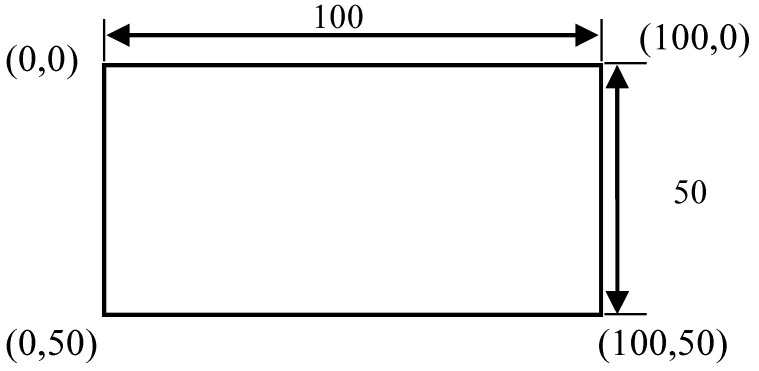
Bounding box output.

**Figure 10 sensors-23-02572-f010:**
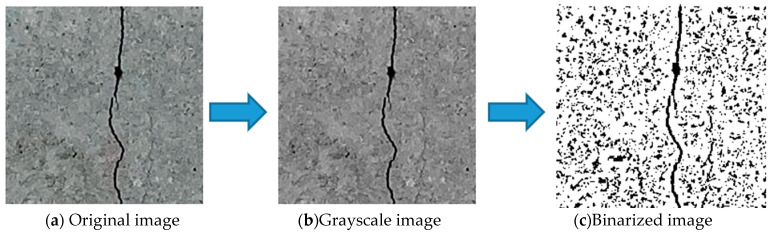
Schematic diagram of the thresholding process.

**Figure 11 sensors-23-02572-f011:**
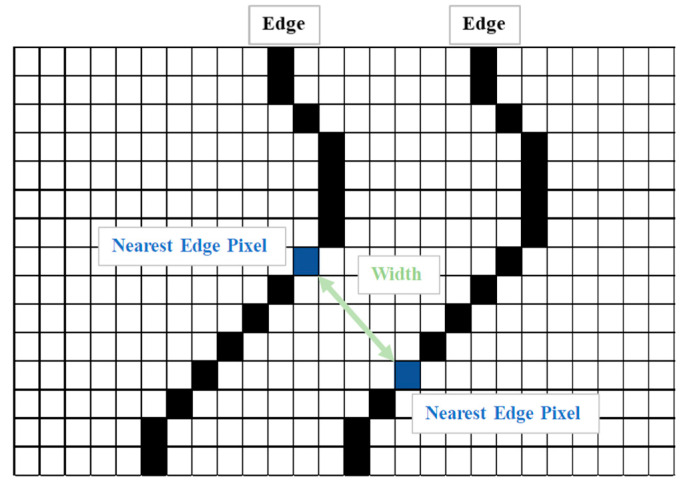
Crack measurement.

**Figure 12 sensors-23-02572-f012:**
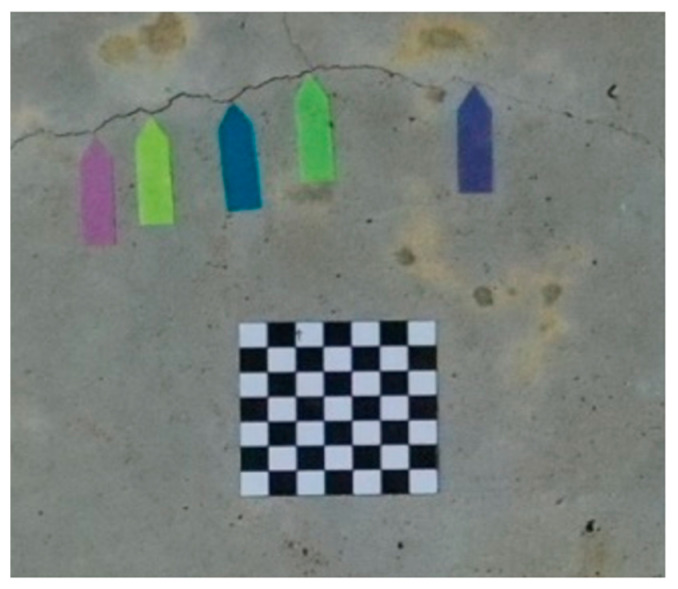
Planar marker.

**Figure 13 sensors-23-02572-f013:**
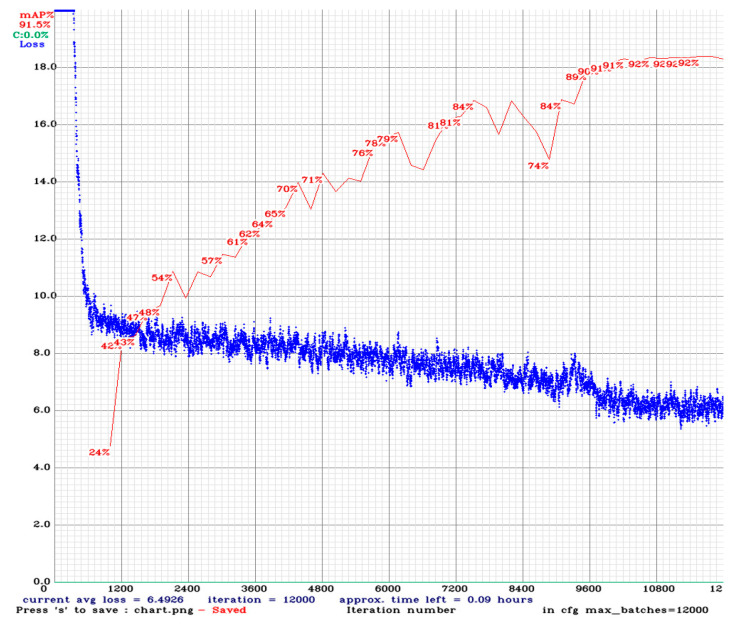
Model training results.

**Figure 14 sensors-23-02572-f014:**
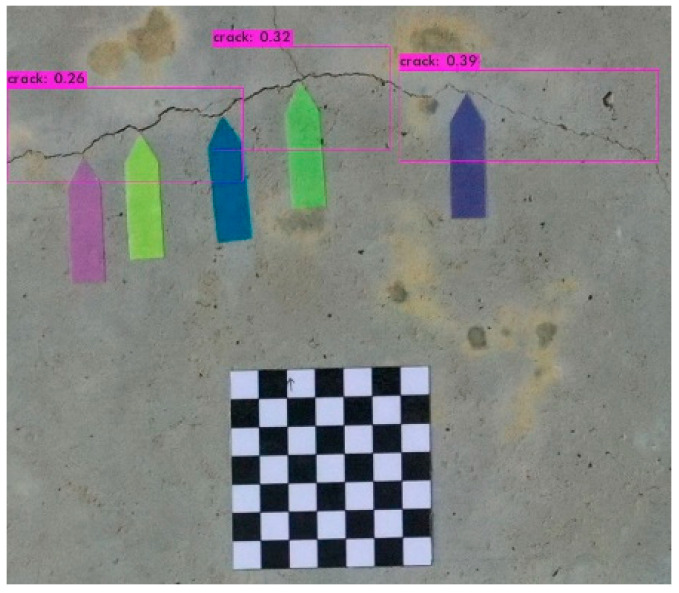
Crack identification outcome.

**Figure 15 sensors-23-02572-f015:**
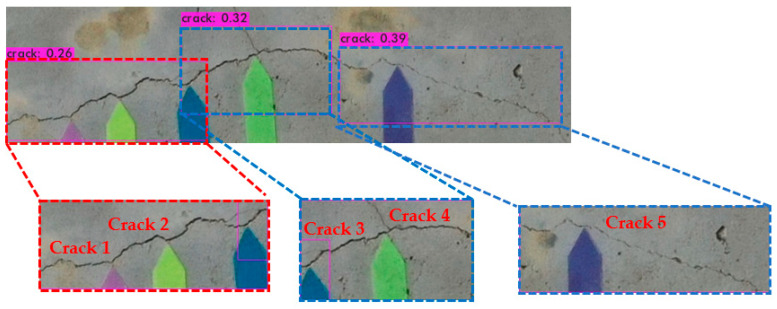
Bounding box cropping.

**Figure 16 sensors-23-02572-f016:**
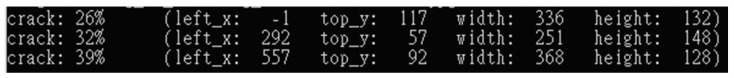
Coordinates and parameters of the bounding boxes.

**Figure 17 sensors-23-02572-f017:**
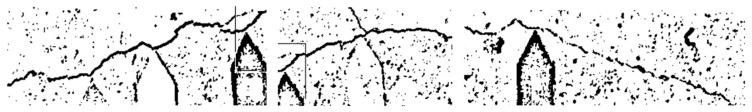
Sauvola local thresholding outcome.

**Figure 18 sensors-23-02572-f018:**
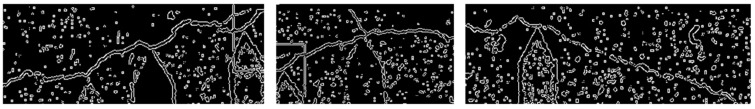
Canny edge detection outcome.

**Figure 19 sensors-23-02572-f019:**
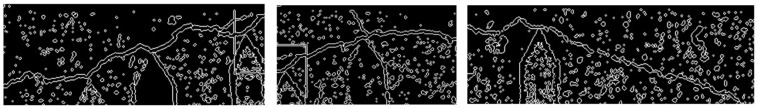
Morphological edge detection outcome.

**Table 1 sensors-23-02572-t001:** Photography equipment [41,42].

Specs	I PHONE12	L1D-20c
Method of Photography	Handheld	Mounted on a DJI Mavic 2 Pro(UAV)
Image Size	1108 × 1478	5472 × 3648
Resolution	96 dip	72 dip
Bit Depth	24	24
Manufacturer	Apple	Hasselblad
Picture	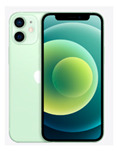	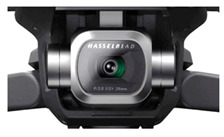

**Table 2 sensors-23-02572-t002:** SDNET2018 images by type [21].

Sample Types	With Cracks (qty.)	Without Cracks (qty.)	Image Size
Bridges	2025	11,595	256 × 256 (96 dpi)
Walls	2608	21,726	
Pavements	3853	14,287	

**Table 3 sensors-23-02572-t003:** Allocation of the training data.

	Training Dataset (80%)	Testing Dataset (20%)	Total
Training images (qty.)	3575	894	4469

**Table 4 sensors-23-02572-t004:** Camera elements of the interior orientation and lens distortion calculation results.

Item	Parameter Name	Parameter	Value
Elements of interior orientation (mm)	Focal length	f	10.856
	Principal point	$x_{0}$	2.688
		$y_{0}$	1.900
Lens distortion	Radial distortion coefficients	$k_{1}$	0.0058
		$k_{2}$	−0.0156
	Tangential distortion coefficients	$p_{1}$	0.0039
		$p_{2}$	0.0004

**Table 5 sensors-23-02572-t005:** Crack identification outcomes for different lighting and backgrounds.

Background: Simple	Background: Complex
Test image 1	Test image 2
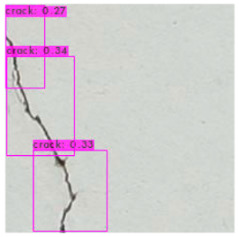	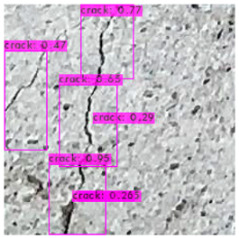
Brightness: 186 (strong light)	Brightness: 191 (strong light)
Test image 3	Test image 4
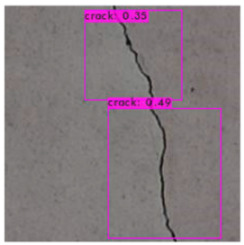	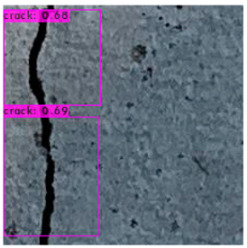
Brightness: 119 (adequate light)	Brightness: 102 (adequate light)
Test image 5	Test image 6
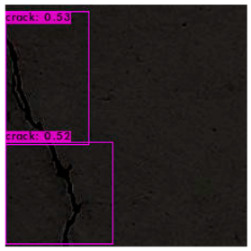	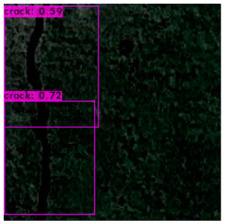
Brightness: 31 (insufficient light)	Brightness: 20 (insufficient light)

**Table 6 sensors-23-02572-t006:** Morphological planar marking measurements (spatial resolution of one pixel: 0.216 mm).

Crack Number	Crack 1	Crack 2	Crack 3	Crack 4	Crack 5
Crack width (in pixels)	2	3	3	3	1
Crack width (mm)	0.432	0.648	0.648	0.648	0.216
Real value (mm)	0.350	0.650	0.700	0.650	0.350
Absolute error (mm)	0.082	0.002	0.052	0.002	0.134
Error rate (%)	23.43	0.31	7.43	0.31	38.29

**Table 7 sensors-23-02572-t007:** Canny planar marking measurements (spatial resolution of one pixel: 0.216 mm).

Crack Number	Crack 1	Crack 2	Crack 3	Crack 4	Crack 5
Crack width (in pixels)	2	3	2	3	1
Crack width (mm)	0.432	0.648	0.432	0.648	0.216
Real value (mm)	0.350	0.650	0.700	0.650	0.350
Absolute error (mm)	0.082	0.002	0.268	0.002	0.134
Error rate (%)	23.43	0.31	38.29	0.31	38.29

**Table 8 sensors-23-02572-t008:** Morphological total station measurements (spatial resolution of one pixel: 0.221 mm).

Crack Number	Crack 1	Crack 2	Crack 3	Crack 4	Crack 5
Crack width (in pixels)	2	3	3	3	1
Crack width (mm)	0.442	0.663	0.663	0.663	0.221
Real value (mm)	0.350	0.650	0.700	0.650	0.350
Absolute error (mm)	0.092	0.013	0.037	0.013	0.129
Error rate (%)	26.29	2.00	5.29	2.00	36.86

**Table 9 sensors-23-02572-t009:** Canny total station measurements (spatial resolution of one pixel: 0.221 mm).

Crack Number	Crack 1	Crack 2	Crack 3	Crack 4	Crack 5
Crack width (in pixels)	2	3	2	3	1
Crack width (mm)	0.442	0.663	0.442	0.663	0.221
Real value (mm)	0.350	0.650	0.700	0.650	0.350
Absolute error (mm)	0.092	0.013	0.258	0.013	0.129
Error rate (%)	26.29	2.00	36.86	2.00	36.86

## Data Availability

Not applicable.

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
