# Peer review of "Combining the YOLOv4 Deep Learning Model with UAV Imagery Processing Technology in the Extraction and Quantization of Cracks in Bridges"

_sensors, 2023, doi:10.3390/s23052572_

Round 1

Reviewer 1 Report

In this study, cracks on bridge surfaces were photographed using a UAV-mounted camera. Using transfer learning, an object detection model was trained, then the model was employed for cracks identification in industrial inspection. The title matches the content, and excellent work has been presented. I recommend the publication of the manuscript after the following improvements is done.

1.      Keywords should be in alphabetical order. (line 23)

2.      Very long paragraph, and it is difficult to follow the main idea of the paragraph (line 43-86)

3.      The conclusion is very brief and does not provide suggestion for the next step (line 391-404).

Reviewer 2 Report

The authors propose an approach for bridge inspection using deep learning to train a model for identifying cracks and photographs captured using a UAV-mounted camera. The paper is well writen, with a few minor errors found and listed as follows.

"its identification accuracy is this low" -> "its identification accuracy is low"

"Figure 3. planar marker" -> "Figure 3. Planar marker"

"CBM then split" -> "CBM then was split"

"images. (Table 4)" -> "images (Table 4)."

"In the identified crack image presented in Figure 13" -> "In the identified crack image presented in Figure 12"

what is the meaning of the percentage in Figure 13?

Some points deserve attention in order to improve paper quality and also the readers' understanding.

For instance, what is the computational load involved in the proposed algorithm? What is the fps of the system?

It is important to compare the obtained results with other works. Why do the authors do not compare their results with the work from [15]? This work seems to be one of the the most similar ones as it also uses YOLO to detect the cracks.

It would be nice to have a general comparison of the results of the proposed work and other approaches. For example, the paper https://www.sciencedirect.com/science/article/pii/S2095809920303301 shows very promising results and accuracy near 100%. What are the advantages of the proposed method compared to already existing ones? A table should be enough to compare advantages and disadvantages of the proposed method with similar work.

Reviewer 3 Report

The submitted manuscript (sensors-2197552) entitled: “Combining the YOLOv4 Deep Learning Model with UAV Imagery Processing Technology in the Extraction and Quantization of Cracks in Bridges”, presents an experimental work that investigates an effective model to identify cracks under uneven lighting or complex component backgrounds of inaccessible concrete bridges or elevated expressway. The paper includes an interesting method to predict the image category, significantly reducing the prediction time and the total calculation amount. The following recommendations must be considered by the authors to improve the overall quality of the manuscript.

1. The authors need to summarize the main points of the abstract and present the methodology more clearly. Please revise and enhance the abstract.

2. The authors didn’t explain the concepts of “UAV-mounted camera” and “Deep learning” in the abstract section, as well as in the introduction. They suggested identifying the terms “UAV Imagery” and “YOLOv4” as referred to the title in the abstract section and in the introduction. Some details could be added, especially for readers unfamiliar with those concepts and terms. 

3. The introduction needs to be strengthened with sufficient background information and a possible future application of this technology. The methodology must be written in more detail for a better understanding.

4. The Research method in the Materials and methods section needs to be enhanced. It is better to add some photos of camera-equipped UAVs with the corresponding explanation. Figure 2 must give a more analytical explanation about “Research process”

5. Figures 3 and 4 should be explained in more detail for a better understanding by readers. Also, add references to previous related experiments. The given information is not sufficient.

6. The results should be compared with justifications supported by related existing literature.

7. The conclusions need to be revised and improved. Please make sure the conclusion section underscores the scientific value added to the paper and the applicability of the findings/results. Furthermore, they need to be more substantial, supported by the results, and more vital information about the experimental methodology and results section.

Based on the above comments, the paper could be suggested to be accepted. However, it is recommended the manuscript must be re-submitted after major revision, providing the suggested improvements.

Reviewer 4 Report

The paper applies the YOLOv4 object detection model for crack detection and some edge detection methods for quantifying cracks. The following comments should be appropriately addressed before the final decision:

1. The language of the paper needs to be improved in terms of grammatical mistakes, capital letters and choice of words. For example, Section 2.1: "... employed to as the data for training ..." and" ...the proposed method proposed" must be corrected.

2. The abstract should be to the point and clear. The current version is long with an unnecessary introduction about the visual inspection.

3. Section 2.4 is somehow confusing. Please write it concisely and concentrate on your method. For example, the authors state that "The preset format is typically 416 × 416, but many scholars use" where they should explain their own method. Please remove the unnecessary explanations.

4. In Section 2.5.1, what do you mean by "Figure 8 depicts the ideal detection method"?

5. The process of thresholding should be further clarified.

6. How did you implement the width measurement in an automated way?

7. The results need to be discussed. For example, Figures 15-17 are left without any explanation. In addition, in these figures, some bounding boxes are seen. It had better remove them.

8. In Tables 6-9, it is suggested to report the error ratios rather than the absolute value. 

9. For making general statements (in the introduction section), please cite the review papers in the field of data-driven and ANN-based SHM rather than specific studies.

Reviewer 5 Report

For abstract:

Shall be more detailed on your research process and emphasize your contribution and try to answer the following questions, such as What are the roles of planar marker and total station in your study? What are the advantages of your method compared with the existing methods? What are the mentioned pixel ratios? What edge detection algorithm was used?

For introduction:

1.      Before heading into Computer Vision, other quantification method shall also be mentioned and compared. And then further illustrate the pros and cons of CV in crack quantification.

2.      Why YOLOv4 model used in this research? Its advantages shall be emphasized with more details.

3.      Review part of Image processing research shall be equivalent with the review part of CV techniques. Too many contents for CV or too little for image processing part. As well, review part shall be concentrated on the research using both CV and image processing techniques. And compared them with yours to emphasise your contributions.

4.      Why planar marker method chosen in this research?Can realize the original function without the marker assistance?

For subsequent part:

1.      Shall be further illustrate the roles played by the two methods: planar marker method and total station method. What are them used for? for being the baseline? What’s the relationship between these two methods and the proposed CV based method?\

2.      For figure 2, the flow chart shall be more detail and sophisticated, try to include more research process and details of the proposed method.

3.      For scale method, its principle shall be introduced first. And then illustrate how is it used in this research.

4.      Segmentation is usually as the next step after bounding box is generated. So what is the difference between Thresholding and Segmentation?

5.      What is the theoretical support of Equation-2 ?

6.      For model training part, more detail shall be added. What platform was used? Pytorch? Tensorflow? What is the configuration? Training batch size?

For Conclusion:

Conclusions firstly shall give a holistic explanation and illustration of the proposed quantification method, and states what is the main contributions and shining point of this paper. After that, prove the effectiveness of the method with some experiment results. Finally, any improvement is needed for the proposed method?

Round 2

Reviewer 3 Report

The revised manuscript sensors-2197552” and title: “Combining the YOLOv4 Deep Learning Model with UAV Imagery Processing Technology in the Extraction and Quantization of Cracks in Bridges” has been revised extensively and improved. The Authors considered all the recommendations and responded to all the criticisms of the previous review successfully, hence, it is suggested to be accepted for publication in its present form.

Care should be taken that Fig. 12 is still missing possibly by mistake.

Author Response

Thank you for agreeing to accept the paper in its present form and recommended it to the editors.

We have updated Fig. 12.

Reviewer 4 Report

The paper is revised and can be considered for publication.

Author Response

Thank you for agreeing to accept the paper in its present form and recommended it to the editors.

Reviewer 5 Report

Overall looks great. The questions proposed in 1st version have been well answered.

Additional question:

The purpose of your research is to quantify the crack where location is hard to get access to. Would the use of a planar marker to determine the scaling factor require inspectors to gain access to the crack location and affix the marker before measuring the dimensions of the crack, and if so, does this pose a conflict with the initial objective of the research?

Author Response

Response : Thank you for pointing this out. 
As you pointed out, the disadvantage of the planar marker method is that it requires the inspector to gain access to the crack location and affix the marker before measuring the dimensions of the crack.

Therefore, this study proposed another method for determining the scale factor (method 2: total station measurement method), which does not require access to the crack location, uses a total station to measure the coordinates of concrete surface features, and then uses these coordinates to calculate the spatial distance , replacing the planar markers. The total station measurement method proposed in this study can eliminate the limitations of the planar marker method and provide a more convenient operating procedure for Quantization of Cracks in Bridges.